# Whole-Genome Sequence of *Aeromonas hydrophila* CVM861 Isolated from Diarrhetic Neonatal Swine

**DOI:** 10.3390/microorganisms8111648

**Published:** 2020-10-24

**Authors:** Toni L. Poole, Wayne D. Schlosser, Robin C. Anderson, Keri N. Norman, Ross C. Beier, David J. Nisbet

**Affiliations:** 1USDA/ARS Plains Area Research Center, 2881 F&B Road, College Station, TX 77845, USA; robin.anderson@ars.usda.gov (R.C.A.); ross.beier@ars.usda.gov (R.C.B.); david.nisbet@ars.usda.gov (D.J.N.); 2USDA/FSIS Plains Area Research Center, 2881 F&B Road, College Station, TX 77845, USA; wayne.schlosser@fsis.usda.gov; 3Department of Veterinary Integrative Biosciences, Texas A&M University College of Veterinary Medicine & Biomedical Sciences, College Station, TX 77843, USA; knorman@cvm.tamu.edu

**Keywords:** aquatic pathogen, food safety, WGS, microbial drug resistance, virulence genes, environmental biome, horizontal gene transfer

## Abstract

*Aeromonas hydrophila* are ubiquitous in the environment and are highly distributed in aquatic habitats. They have long been known as fish pathogens but are opportunistic human pathogens. *Aeromonas* spp. have persisted through food-processing safeguards and have been isolated from fresh grocery vegetables, dairy, beef, pork, poultry products and packaged ready-to-eat meats, thus providing an avenue to foodborne illness. A beta-hemolytic, putative *Escherichia coli* strain collected from diarrheic neonatal pigs in Oklahoma was subsequently identified as *A. hydrophila*, and designated CVM861. Here we report the whole-genome sequence of *A. hydrophila* CVM861, SRA accession number, SRR12574563; BioSample number, SAMN1590692; Genbank accession number SRX9061579. The sequence data for CVM861 revealed four *Aeromonas*-specific virulence genes: lipase (*lip*), hemolysin (*hlyA*), cytonic enterotoxin (*ast*) and phospholipid-cholesterolacyltransferase (GCAT). There were no alignments to any virulence genes in VirulenceFinder. CVM861 contained an *E. coli* resistance plasmid identified as IncQ1_1__M28829. There were five aminoglycoside, three beta-lactam, and one each of macrolide, phenicol, sulfonamide, tetracycline and trimethoprim resistance genes, all with over 95% identity to genes in the ResFinder database. Additionally, there were 36 alignments to mobile genetic elements using MobileElementFinder. This shows that an aquatic pathogen, rarely considered in human disease, contributes to the resistome reservoir and may be capable of transferring resistance and virulence genes to other more prevalent foodborne strains such as *E. coli* or *Salmonella* in swine or other food production systems.

## 1. Introduction

*Aeromonas* spp. comprise numerous species that are widely distributed in aquatic environments ranging from sewage, marine and freshwater to water treatment plants [1,2,3]. They may form biofilms in pipes and are considered potential emerging pathogens of water by the Environmental Protection Agency (EPA). They are ubiquitous in the environment having been isolated from far-ranging non-aquatic niches [4,5]. *Aeromonas* spp. have persisted through food-processing safeguards and have been isolated from fresh grocery vegetables, dairy, beef, pork, poultry products, and packaged ready-to-eat meats [6,7]. Some believe food-producing animals have contributed to human dissemination [8]. *Aeromonas* spp. are opportunistic pathogens and have been identified as the etiologic agent in a variety of infections in immunocompetent and immunocompromised humans [4,9]. Severe infections include septicemia, peritonitis, endocarditis, gastroenteritis, wound infections, necrotizing fasciitis, and hemolytic-uremic syndrome (HUS) [4,10,11,12]. *A. hydrophila* may encode a number of virulence factors including hemolysins, aerolysins, adhesins, enterotoxins phospholipase, and lipase [9,11,13].

*Aeromonas* spp. are Gram-negative bacilli that are oxidase and catalase positive. They are distinguishable based on temperature growth traits [8]. One group is mesophilic, displaying motility, and is associated with human disease. This group is typified by *A. hydrophila* and displays optimal growth between 35 °C and 37 °C. Another group is psychrophilic, non-motile, and is associated with fish diseases. This group is typified by *A. salmonicida* and displays optimal growth between 22 °C and 25 °C [5]. Different metabolic categories of growth and abilities to colonize many niches with the potential for disease define *Aeromonas* spp. as transboundary pathogens. 

Aquaculture increasingly provides a sustainable, safe alternative to more traditional livestock and poultry production as a food source. The amount of fish consumption coming from aquaculture compared to capture has increased over the years. For the time periods 1986–1995, 1996–2005, 2006–2015, and 2016–2018 the average contributions of fish production from aquaculture were 14.6%, 27.2%, 39.9%, and 46.0%, respectively [14]. The increase in farm fisheries for food production has correspondingly led to an increase in antimicrobial use [15]. Norway, however, has developed a combination of aquaculture farming and regulatory practices that have successfully mitigated much of the need for antimicrobials [16]. Part of the governance included use of zoo sanitary methods and vaccines [16]. 

Traditionally, clinical and agriculture settings have been the primary the focus of public health officials for monitoring antimicrobial-resistant microorganisms. Currently, a wider scope includes the environmental microbiome, mobilome, and resistome. A collaborative effort by many countries to address antimicrobial resistance has been termed “One Health”. One Health is an effort by health professionals to minimize antimicrobial resistance in human and veterinary medicine, agriculture, aquaculture, environmental niches and molecular genetic reservoirs [17,18]. To accomplish this effort, it is necessary to elucidate reservoirs of microflora and their respective resistance and virulence profiles. 

Whole-genome sequencing and metagenomic next-generation sequencing have provided insight into the complex antimicrobial resistance reservoirs that exist. A horizontal flow of antimicrobial resistance and virulence genes occurs due to mobile genetic elements such as plasmids, transposons, integrons and integrative and conjugative elements (ICEs) that can acquire resistance and virulence genes as they move between bacteria. 

*A. hydrophila* CVM861 was chosen for WGS to obtain the complete genotype including virulence genes that *A. hydrophila* intrinsically possesses. This potentially elucidates the possible fluidity via recombination between chromosomal and plasmid locations of virulence and resistance genes. It also provides information on the resistome of food production animals and the larger global resistome.

## 2. Materials and Methods

### 2.1. Bacterial Isolate

*Aeromonas hydrophila* CVM861 was previously identified and found to possess the macrolide inactivation gene cluster *mphA-mrx-mphR.* The EMBL GenBank accession sequence for the previously characterized sequence of 9948bp is AY522923. The GenBank BioSample accession number for the WGS of CVM861 is SAMN15950692. The BioProject number is PRJNA660480.

### 2.2. Antimicrobial Susceptibility Testing

Susceptibility testing was performed by broth microdilution according to Clinical Laboratory Standards Institute (CLSI) methods [19] on CVM861. A Sensititre^®^ automated antimicrobial susceptibility system was used according to the manufacturer’s instructions (Trek Diagnostic Systems, Westlake, OH, USA) using the National Antibiotic Resistance Monitoring System (NARMS) panel CMV3AGNF (Trek Diagnostic Systems). The following antimicrobials were tested: amoxicillin/clavulanic acid, ampicillin, cefoxitin, ceftiofur, ceftriaxone, chloramphenicol, ciprofloxacin, gentamicin, nalidixic acid, streptomycin, sulfisoxazole, tetracycline, and trimethoprim/sulfamethoxazole. *E. coli* ATCC 25922, *E. coli* ATCC 35218, and *Enterococcus faecalis* ATCC 29212 were used as quality control organisms. Data were interpreted using CLSI break points unless unavailable, and then break points from the NARMS 2010 annual report were used [20]. Minimum Inhibitory Concentrations (MICs) for rifampicin were determined manually by broth microdilution using previously described methods [21]. 

### 2.3. Whole-Genome Sequencing (WGS)

DNA purification employed the QIAamp 96 DNA QIAcube HT kit (Qiagen, Valencia, CA, USA). DNA quality and quantity were assessed by absorbance and fluorescence using a FLUOstar Omega plate reader (MG Labtech, Cary, NC, USA). DNA was sent (Novogen Corporation, INC. Sacramento, CA, USA) for WGS on an Illumina MiSeq^®^ sequencer (150-bp paired-end reads). Sequences were uploaded to PATRIC for assembly [22]. Assembly was performed on PATRIC twice using both the Unicycler and MetaSpades (v3.12.0) options. The ResFinder [23], VirulenceFinder (2.0 database) [24] PlasmidFinder (2.1 database) and MobileElementFinder (1.0.2.0 database) databases from the Center for Genomic Epidemiology (CGE) (Danish Technical University, Lyngby, Denmark) available at https://cge.cbs.dtu.dk/services/MobileElementFinder/ were downloaded for sequence alignments. The alignments were performed with MagicBlast [25]. *A. hydrophila* CVM861 contigs were also aligned with *Aeromonas*-specific virulence genes in Genbank using Magic-Blast [25].

## 3. Results

The sequences generated from the WGS were submitted to Genbank as reads. The Genbank accession information was: SRA accession number, SRR12574563; BioSample number, SAMN1590692; BioProject number PRJNA660480; Genbank accession number SRX9061579.

Assembly with the Unicycler option produced 92 contigs with a genome of 4800554 bp, a GC content of 61.45% and N50 of 132950. The assembly using the Metaspades option produced 261 contigs with a total length of 4933241 bp, and GC content of 61.42% and an N50 of 130230. 

There were no alignments to 2538 virulence gene sequences provided by CGE. Therefore, ten *Aeromonas*-specific virulence gene sequences from Genbank were aligned (Table 1). Four *Aeromonas*-specific virulence genes were identified in the CVM861 genome: lipase (*lip*), hemolysin A (*hlyA*), cytonic enterotoxin (*ast*) and phospholipid-cholesterolacyltransferase (GCAT). Lipase had a 94% identity. The other three had a >95% identity.

CVM861 aligned to 13 sequences in the ResFinder database (3100 gene sequences), all of the alignments had a >95% identity (Table 2). Correspondingly, CVM861 displayed phenotypic resistance to ampicillin, amoxicillin/clavulanic acid, azithromycin, gentamicin, streptomycin, sulfisoxazole, tetracycline, trimethoprim/sulfamethoxazole and high-level resistance to erythromycin (≥512) [26]. However, the only unexpected result was phenotypic susceptibility to chloramphenicol.

Comparison of CVM861 to the PlasmidFinder database (469 gene sequences) resulted in the alignment to a single plasmid, IncQ1 (identified as IncQ1_1__M28829 in the database). The CVM861 sequence fragment displayed a 100% alignment with an IncQ *Escherichia coli* plasmid RSF1010 sequence that was in the Enterobacteriaceae database. 

CVM861 was aligned to the MobileElementFinder database version v1.0.2 (9 June 2020) containing 4452 elements (Table 3). When both assembly methods were combined, there was alignment to 20 transposons (Tn), four *Salmonella* genomic island 1 sequences (SGI1), four each: insertion sequences (IS), and Intergrative and conjugative elements (ICEs). Overall, there were alignments to 34 different potential mobile elements.

## 4. Discussion

*Aeromonas* infections can be difficult to control for many reasons. They have inducible chromosomal β-lactamase resistance, primarily to ampicillin [4,26]. Chromosomally located resistance genes are not easily transferred horizontally to other bacteria and are considered intrinsic to the organism. However, numerous mobile DNA elements such as, ISs, ICEs, transposons, integrons, and incompatibility plasmids exist that allow resistance and virulence genes to transfer horizontally between bacterial species [27,28,29,30]. Incompatibility plasmids transferred by conjugation between bacterial cells are the primary source of resistance gene dissemination [30]. Conjugation between Aeromonads and other genera has been demonstrated [31,32]. Whole-genome sequencing further elucidates the global dissemination of antimicrobial resistance, virulence, heavy metal resistance and other genetic determinants that may become located on mobile genetic elements. 

Bacterial cells can possess multiple plasmids, of varying size, without substantially affecting their growth [33]. In this study CVM861 possessed a single incompatibility plasmid, IncQ1. IncQ plasmids are among those, IncA/C, IncU, IncQ, IncF, IncI, and ColE-type, that have been previously identified in *Aeromonas* spp. [27]. All of those plasmids were from *Aeromonas* spp. isolated from fish or the environment. However, they are of concern because they are broad-range plasmids that are often found in clinical or animal environments [27]. An *Aeromonas* sp. isolated from the Ter river in Spain was fully sequenced in 2012 and found to carry IncU plasmid (pP2G1) [34]. The genetic map for the pP2G1 produced a total sequence length of 26,645 bp and possessed the MphA gene cluster similar to CVM861. 

Macrolide phosphotransferase (MphA) is an enzyme that phosphorylates erythromycin and other macrolide antibiotics; thereby, inactivating their antibiotic properties. Phosphorylation confers high-level resistance to macrolides. An MphA gene cluster (*mphA-mrx-mphR)* was first found in *E. coli* [35]. Our laboratory was screening a number of *E. coli* isolates for *mphA-mrx-mphR* in 2005. It was discovered in CVM861 which was then determined to be *Aeromonas hydrophila* and not *E. coli* [26]. The MphA gene cluster was partially mapped and 9948 bp were sequenced by our laboratory [26]. At that time, no incompatibility plasmid was identified in the isolate. Several resistance genes were located upstream of the MphA gene cluster in a class 1 integron: *aad*A2, *sul*1 and *dfr*A. The *aph(3′)-Ia* gene was located downstream of the MphA gene cluster. All were downstream of a Tn*21* transposon [26]. 

High-level resistance to macrolides is not necessary for Gram-negative bacteria, such as *E. coli* or *Aeromonas* spp., due to efflux pumps that transport macrolide antimicrobials out of the cell. However, it demonstrates how dissemination of resistance genes occurs when they are physically linked on mobile genetic elements that can be transferred to other genera or strains of microorganisms [36,37].

An unidentified bacterial spp. was collected from a municipal sewage treatment plant in Germany [38]. Conjugative kanamycin resistance plasmids were selected via transformation into *E. coli*. The plasmid characterized, pRSB225 was an IncF plasmid that displayed resistance to the same categories of antimicrobials as CVM861 including high-level resistance to macrolide antimicrobials [38]. Resistance genes on pRSB225 were linked to genes coding for mobile genetic elements, primarily insertion sequences and transposases [38]. 

Overall, CVM861 produced genomic alignments that were anticipated for a multidrug-resistant strain. Initially, alignment to seven antimicrobial classes from ResFinder corresponded with the phenotypic susceptibility profile with the exception of chloramphenicol. The *cat*B3 gene identified in CVM861 encodes chloramphenicol acetyltransferase that catalyzes the acetylation of the 3′-OH of chloramphenicol and provides one mechanism for resistance to phenicol antimicrobials [39]. Resistance to chloramphenicol may also be mediated by efflux [40,41]. However, the CVM861 MIC for chloramphenicol was 8 µg/µl, which is indicative of a susceptible strain. This suggests a gene mutation or other circumstance exists to prevent the production of a functional protein. The case sometimes exists that genotypic indicators of microbial resistance do not necessarily translate into phenotypic expression.

The lack of virulence genes in CVM861 to those in Enterobacteriaceae suggests that none had been translocated to the IncQ plasmid. 

## 5. Conclusions

Although *Aeromonas* is a rarely considered a human pathogen, it still contributes to the resistome reservoir and may be capable of transferring resistance and virulence genes to other more prevalent foodborne strains such as *E. coli* or *Salmonella* in swine or other food production systems. The ability to provide comprehensive knowledge of the entire genome including mobile genetic elements allows elucidation into the global dissemination of these elements. It is important to continue studies of this type for both health and food safety management.

## Figures and Tables

**Table 1 microorganisms-08-01648-t001:** Alignment of *Aeromonas hydrophila* CVM861 with Genbank sequences for *Aeromonas-*specific virulence genes.

Gene	Genbank Number	Base Pairs	% Alignment
Aerolysin (*aer*)	HQ425626.1	1482 complete gene	0
Flagellin (*fla*)	JN215209.1	459 partial gene	0
	JN215210.1	429 partial gene	0
Lipase (*lip*)	AB237183.1	2418 complete gene	94.086
Hemolysin (*hlyA*)	U81555.1	2265 complete gene	95.85
Cytonic enterotoxin (alt)	L77573.1	1371	0
Lateral flagella (*lafK*)	DQ650654.1	1377 complete gene	0
Cytonic enterotoxin (*ast*)	AF419157.1	4623 complete gene	96.19
Phospholipid-cholesterolacyltransferase (GCAT)	GQ856318.1	1155 complete gene	98.614
*ahpB* (elastase)	AF193422	1929 complete gene	0

**Table 2 microorganisms-08-01648-t002:** Alignment of *Aeromonas hydrophila* CVM861with sequences from the ResFinder database.

ResFinder Database	ResFinder Gene Reference	% Identity	Protein ID	Protein Definition
aminoglycoside	*aad*A1_3_JQ414041	100.0	AFI58053.1	aminoglycoside 3′-adenyltransferase (*Pseudomonas aeruginosa)*
	*aph*(3′)-Ia_1_V00359	100.0	CAA23656.1	MULTISPECIES: aminoglycoside O-phosphotransferase APH (3′)-Ia (Bacteria)
	*aad*A2_1_NC_010870	99.9	WP_001206356.1	MULTISPECIES: ANT (3′′)-Ia family aminoglycoside nucleotidyltransferase AadA2 (Bacteria)
	*ant*(3′′)-Ia_1_X02340	99.8	CAA26199.1	put. AAD (3)(9) precursor (plasmid) (*Escherichia coli)*
	*aac*(3)-IV_1_DQ241380	97.2	ABB43029.1	aminoglycoside acetyltransferase (*Escherichia coli*)
beta-lactam	*amp*H_2_HQ586946	98.2	AEF33353.1	adenylate kinase beta-lactamase (*Aeromonas hydrophila*)
	*cph*A1_7_X57102 ^a^	95.3	CAA40386.1	beta-lactamase (*Aeromonas hydrophila*)
	*cph*A2_6_AHU60294 ^a^	95.3	AHU60294.1	citrate lyase subunit beta (*Salmonella enterica* subsp. Enteric serovar Enteritidis str. EC20121812)
macrolide	*mph*(A)_2_U36578	99.7	AAB03644.1	macrolide phosphotransferase K (*Escherichia coli*)
phenicol	*cat*B3_2_U13880	100.0	AAA90938.1	chloramphenicol acetyltransferase (plasmid) ((*Enterobacter*) *aerogenes*)
sulfonamide	*sul*1_5_EU780013	99.9	ACF06160.1	dihydropteroate synthase (plasmid) (*Klebsiella pneumoniae*)
tetracycline	*tet*(E)_3_CP000645	100.0	ABO92308.1	tetracycline resistance protein TetA(E) (plasmid) (*Aeromonas salmonicida* subsp. salmonicida A449)
trimethoprim	*dfr*A12_8_AM040708	100.0	CAJ13564.1	dihydrofolate reductase (*Escherichia coli*)

a—cphA1_7_X57102 and cphA2_6_AHU60294 aligned at same location.

**Table 3 microorganisms-08-01648-t003:** Alignment of *Aeromonas hydrophila* CVM861with sequences from the MobileElementFinder database.

	Unicycler Assembly		MetaSpades Assembly	
MobileElementFinder Reference	Number of Alignments *	Maximum Alignment	Number of Alignments	Maximum Alignment
MGIVchHai6|1|AXDR01000001	1	100.00	1	100.00
PGI1-PmPEL|1|KF856624	1	100.00	1	100.00
SGI1-PmGUE|1|JX121641	1	100.00	1	100.00
SGI1-PmVER|1|JX121640	1	100.00	1	100.00
SGI1-V|1|HQ888851	1	100.00	1	100.00
Tn*1681*|1|L36547.1	1	100.00	1	100.00
Tn*2610*|1|AB207867.1	1	100.00	1	100.00
Tn*2670*|1|AP000342.1	1	100.00	1	100.00
Tn*4352*|1|M20306.1	1	100.00	1	100.00
Tn*6026*|1|GQ150541	2	100.00	2	100.00
Tn*6234*|1|HG934082	1	100.00	1	100.00
Tn*6284*|1|KU254577	1	100.00	1	100.00
Tn*6285*|1|KX646543	1	100.00	1	100.00
TnAs3|1|CP000645	1	100.00	1	100.00
PGI2|1|MG201402	2	100.00	1	100.00
SGI1-Pm2CHAMA|1|MF372716	1	100.00		
Tn*5045*|1|FN821089.1	1	100.00		
Tn*6027*|1|HQ840942	1	100.00		
Tn*6060*|1|GQ161847	1	100.00		
Tn*6112*|1|HQ423158	1	100.00		
Tn*6162*|1|JF826498	1	100.00		
Tn*6249*|1|LK054503	1	100.00		
ICEKkKWG1|1|LN869922	2	100.00		
Tn*21*|1|AF071413.3	1	99.98	1	99.98
Tn*6016*|1|KC543497	1	99.97	1	99.97
IS5|1|J01735	1	99.75	1	97.63
ISAs34|1|CP000644	1	99.14		
ICE(Tn4371)6067|1|CP000884	2	98.79	2	98.79
ISAhy1|1|CP000462	3	97.29	1	97.29
ICEPmiChn3|1|KY437727	1	97.08		
ISAhy2|1|FM877486	1	95.08	1	95.08
ICEEcoUMN026-1|1|CU928163			1	99.34
Tn*6179*|1|KX011025			1	99.88
Tn*6180*|1|KX011025			1	99.88
Tn*6279*|1|KT317075			3	99.88

* Multiple alignments; means that parts of the gene aligned in more than one contig. The maximum alignment from any one contig is reported.

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
