# Peer review of "Whole-Genome Sequence of Aeromonas hydrophila CVM861 Isolated from Diarrhetic Neonatal Swine"

_microorganisms, 2020, doi:10.3390/microorganisms8111648_

Round 1

Reviewer 1 Report

Letter to Authors
microorganisms-968730-v1
Whole genome sequence of Aeromonas hydrophila CVM861 isolated from diarrhetic neonatal swine
Toni L. Poole, Wayne D. Schlosser, Robin C. Anderson, Keri N. Norman, Ross C, Beier, David J. Nisbet

201004

Dear authors,
You have sequenced and characterized an Aeromonas strain carrying drug resistance genes. You have presented the problem, methods, and your findings well. All of the tables are appropriate and necessary. Citations are well-balanced and sufficient. This work is thus appropriate to readers of this journal.
You should, however, make some revision and corrections on the current delivery before publication. Story flow of the introduction section is not very good. See below for detail. I am looking forward to seeing your paper published in the journal.

L2
Aeromonas hydrophila -> in Italics

L31 keywords
Aeromonas hydrophila, Whole Genome Sequencing -> delete and replace
Do not list words which appear also in the title. Duplicate hits upon computer search do not make sense. Give words that do not appear in the title to draw attention from wider readership. Posting words that neither appear in the abstract is better, because even in full-text search/indexing robots may not weigh much on words deeper (posterior) in the text. Hint: zoonosis, environmental microbiome, antibiotics, endotoxin, horizontal gene transfer, etc. WGS is good.

L48-81
Story flow over here is not very good driving readers from side to side.

L48
Somewhat redundant or not well linking with the previous paragraph.

L55
Consider omitting this paragraph. This is not essential.

L61,69
These general statements are usually put forward to draw attention from wider readership. Which would be put forward depends on what you focus on.

L69 This paragraph interferes two paragraphs talking about drug resistance.

L74
Why anti-macrolide only here? See Table 2.

L82,88
Where was the strain from should be mentioned in brief either of these paragraphs.

L110
PATRIC, Unicycler, MetaSpades
References or citation of URLs are necessary.

L110-114
Assembly was done .. for sequence alignments.
You are talking about two different procedures: assembling and annotation of the assembly. Break sentence into these matters.

L113
available -> available at

L115-116
MagicBlast -> Magic-Blast
See L278.

L123
Genbank accession number SRX9061579
Which assembly did you deposit? Did you compare and combine two assemblies given by Unicycler and MetaSpades?

L127
>94% identity ??
identity with known sequences?

L131
Break sentence.

L148
Delete excess horizontal line at the bottom.

L157
Aeromonads -> {aeromonads (English), Aeromonas (in Italics) (Latin)}

L173
spp. -> species

L192 conclusions
Delete this section for comciseness.

L210 references
Check the reference list carefully again from the beginning. Reference lists are frequently den of errors. You might add, omit or swap citation in the main text on the way internal revision. Numbering of the references might then shift. If so, readers think you are making irrelevant citation. It is the authors' responsibility that all references are properly cited.

L212,215,etc (many)
Make sure if dot is necessary for appreviated journal title word. Consult with the layout editor.

L214
Colon in Roman.

L237,239,etc
Aeromonas -> in Italics
Abbreviated journal title?

L295
Micro. Drug Resis. -> Microb. Drug Resist. ?

Author Response

Dear reviewer,

Thank you for your rapid review of our manuscript.

The title and key words have been fixed. Changes have been to the introduction to improve readability. More changes would have been made; however, the other reviewer was pleased with the introduction.

The paragraph on macrolide resistance was moved to the discussion. This required a few other changes that have been tracked throughout the manuscript.

The PATRIC reference was added. Unicycler and MetaSpades are found within PATRIC.

L110-114 was a punctuation problem and was fixed with the addition of a period.

Available at and Magic-Blast were also fixed.

Neither assembly was deposited to Genbank. The sequence was deposited as reads. That is described in the first sentence of the results.

The other changes were made and line 192 was deleted.

The references were all checked and one was added.

Reviewer 2 Report

This is a well performed study. The Introduction is ok, the results are well presented and the very short conclusion summarises the results obtained.

Minor points:

Line 59+60: The increase in farm fisheries for food production has correspondingly led to an increase in antimicrobial use (14). I agree to this statement in many ways, but there are positive reports as well: See this reference: Midtlyng Pj, Grave K, Horsberg TE. What has been done to minimize the use of antibacterial and antiparasitic drugs in Norwegian aquaculture? Aquaculture Research 2011, 42, 28-34.

By the introduction of vaccines, the Norwegian aquaculture industry has reduced its use of antibiotics by 99% of trout and salmon, compared to the 1987 level.

L 182+183: CVM863 encodes chloramphenicol acetyltransferase. That catalyzes the acetylation of the 3´-OH of chloramphenicol and provides one mechanism for resistance to phenicol antimicrobials; change to: CVM863 encodes chloramphenicol acetyltransferase that catalyzes the acetylation of the 3´-OH of chloramphenicol and provides one mechanism for resistance to phenicol antimicrobials

L193: Although Aeromonas is a rarely considered a human pathogen; change to: Although Aeromonas is a rarely considered human pathogen

Author Response

The co-authors and I thank the reviewer for their comments, particularly for the mention of the Midtlyng Pj, Grave K, Horsberg TE article. This provides an example of agriculture significantly decreasing use of antimicrobials. This was added to the introduction.

Line 182-183 now 227 was changed

Line 193 now line 238, issue was changed.